# Determinants of Post-COVID-19 Conditions among SARS-CoV-2-Infected Patients in Saudi Arabia: A Web-Based Cross-Sectional Study

**DOI:** 10.3390/diseases10030055

**Published:** 2022-08-23

**Authors:** Mohammed Samannodi, Hassan Alwafi, Abdallah Y. Naser, Abdullah A. Al Qurashi, Jumanah T. Qedair, Emad Salawati, Mohammed A. Almatrafi, Rakan Ekram, Rahaf I. Bukhari, Maryam Dahlawi, Bayan Hafiz, Roaa Mandora, Ranin K. Alsindi, Sarah Tayeb, Faisal Minshawi

**Affiliations:** 1Department of Medicine, Faculty of Medicine, Umm Al-Qura University, Mecca 24382, Saudi Arabia; 2Faculty of Medicine, Umm Al-Qura University, Mecca 24382, Saudi Arabia; 3Al-Noor Specialist Hospital, Mecca 24241, Saudi Arabia; 4Department of Applied Pharmaceutical Sciences and Clinical Pharmacy, Faculty of Pharmacy, Isra University, Amman 11622, Jordan; 5College of Medicine, King Saud bin Abdulaziz University for Health Sciences, Jeddah 22384, Saudi Arabia; 6King Abdullah International Medical Research Center, Jeddah 22384, Saudi Arabia; 7Department of Family Medicine, Faculty of Medicine, King Abdulaziz University, Jeddah 22254, Saudi Arabia; 8Department of Pediatrics, Umm Al-Qura University, Mecca 24382, Saudi Arabia; 9School of Public Health and Health Informatics, Umm Al-Qura University, Mecca 24382, Saudi Arabia; 10Department of Medicine, King Fahad Armed Forces Hospital, Jeddah 23311, Saudi Arabia; 11Department of Laboratory Medicine, Faculty of Applied Medical Sciences, Umm Al-Qura University, Mecca 24382, Saudi Arabia

**Keywords:** novel coronavirus, acute COVID-19, post COVID-19 symptoms

## Abstract

**Background:** Large numbers of people infected with COVID-19 developed acute symptoms. Post-COVID-19 conditions have been reported after recovery or discharge from the hospital. However, little is known about the prevalence and possible risk factors of post-COVID-19 conditions in the Saudi community. Here, we describe the incidence of post-COVID-19 conditions among the general population of Saudi Arabia. **Methods:** We conducted a cross-sectional, nationwide study using an online survey in Saudi Arabia from 1 September 2021 to 28 February 2022. The survey was distributed using social media platforms, such as Twitter, WhatsApp, and Facebook. Patients with SARS-CoV-2 infection were included in the questionnaire adapted from published studies. **Result:** The study enrolled 7520 individuals who were previously infected with SARS-CoV-2. Most patients in our study were symptomatic and their acute symptoms may persist for more than six days. On the other hand, long-term complications may develop and continue for an extended period (post-COVID-19 conditions). Most of these complications are respiratory, neurological, psychological, or skin related. The proportion of long-term complications reported in this study is 36% among SARS-CoV-2-infected individuals. In addition, being female, old age, number of chronic complications, long-term medication, length of stay in hospital and intensive care unit, and duration of acute symptoms may be significant predictors of post-COVID-19 symptoms. **Conclusion:** In conclusion, the incidence of post-COVID-19 conditions among the Saudi population was high, which urges further investigation into the risk factors associated with post-COVID-19 symptoms.

## 1. Introduction

SARS-CoV-2 has been a significant burden and threat to human health and public safety [1,2]. Individuals affected by SARS-CoV-2 demonstrate clinical symptoms after 5–6 days of incubation, which mainly differs based on the individual’s age and immunity [2,3] The most common clinical symptoms of patients with active infection are cough, fever, fatigue, sputum production, shortness of breath, and loss of taste or smell. However, less commonly seen clinical symptoms are headache, muscle weakness, breathlessness, sore throat, and pleuritic pain. Individuals infected with SARS-CoV-2 are likely to suffer from mild to moderate illness and usually recover without complications or hospitalization [1,2,3] However, COVID-19 sometimes causes prolonged symptoms (lasting weeks to months), even though the infection has subsided. These are the “Long-term effects of Coronavirus” (Post-COVID-19 Conditions), which are defined as the persistence of symptoms or development of consequences after four weeks from the diagnosis of SARS-CoV-2 infection [4] The symptoms of post-COVID-19 conditions include (but are not limited to): extreme tiredness, shortness of breath, and depression. Moreover, most people recover from post-COVID-19 conditions within 12 weeks [5,6,7]. In China, a study by Huang et al. found that COVID-19 survivors were mainly troubled with fatigue or muscle weakness, sleep difficulties, and anxiety or depression [8]. Furthermore, it showed that post-COVID-19 conditions were prevalent among patients with a median age of 57.0 (IQR 47.0–65.0) years and more among men [8]. In France, Garrigues et al. reported that 41.7% of post-COVID-19 individuals suffer from dyspnea, 55% suffer from fatigue, 30.8% suffer from sleep disturbances, and only 10.8–13.3% suffer from loss of taste and smell [9]. Additionally, in a prospective cohort study from Wuhan, China, most post-COVID-19 patients (76%) reported the presence of at least one chronic symptom and most of the symptoms were similar to the previous studies [10]. The exact causes of post-COVID-19 conditions are still being investigated. Some of the reasons (mainly psychological) may be due to treatment in the intensive care unit (ICU) [4]. However, in a comprehensive review that included 98 articles, it has been reported that long COVID-19 can be caused by organ damage due to acute-phase infection [11]. According to the Centre for Disease Control and Prevention (CDC), around one in five (19%) American adults who have had COVID-19 have “Long COVID conditions [12]. In the United Kingdom (UK), the Office for National Statistics (ONS) reported that around 2 million (3%) of the UK population are suffering from self-reported long-COVID symptoms. However, in Saudi Arabia, few studies have investigated the prevalence of long-COVID conditions. Herein, our study aims to describe the prevalence and symptoms of post-COVID-19 conditions in the Saudi population.

## 2. Methods

### 2.1. Study Design

This study was a cross-sectional, nationwide study using an online survey conducted in Saudi Arabia from July to September 2021.

### 2.2. Sampling Strategy

According to the Raosoft sample size calculation, the estimated number of participants is 384 [13]. COVID-19 patients from the northern, southern, eastern, western, and central regions of Saudi Arabia were included. Only those who were SARS-CoV-2 infected were included in the analysis. We used an online link published on social media platforms such as Facebook, Twitter, and WhatsApp to distribute the questionnaire to the Saudi community. It was stated that no identifier information was asked, nor were there positive or negative consequences upon accepting or refraining from filling in the questionnaire.

### 2.3. Study Tool

In light of the available literature, a self-report questionnaire was adopted from previously published studies, querying demographics, data on getting infected with COVID-19, and post-COVID-19 persistent symptoms [14,15]. After structuring the questionnaire using a Google Survey, data were collected for three months, from 1 September 2021 to 28 February 2022. Data were kept safe with authorized access only and Institutional Review Board (IRB: HAPO-02-K-012-2021-08-708) approval on 8 August 2021 was obtained from Umm Al-Qura University (UQU).

After data cleaning and transformation from Excel format (Microsoft, Redmond, WA, USA), statistical analysis was performed with SPSS Questions targeting data on getting infected with acute COVID-19 and post-COVID-19 symptoms involving 20 multiple choice and multiple-answer questions including primary demographic data, such as their age, ethnicity, body mass index, alcohol habit, number of comorbid disorders, type of comorbid disease they have without recording any identifying data for confidentiality.

Finally, the survey included questions about the acute phase of COVID-19 infection, such as the severity, symptoms, hospital admissions, and treatments. It also had questions about post-COVID-19 symptoms such as the number of complications, type of complication, and the mean duration since the onset of the symptoms. For further details on the questionnaire, refer to the Appendix A.

### 2.4. Statistical Analysis

Descriptive statistics were used to describe patients’ demographic characteristics. Continuous data were reported as mean ± SD. Categorical data were reported as percentages (frequencies). Logistic regression was used to identify predictors of persistent post-COVID-19 symptoms. A two-sided *p* < 0.05 was considered statistically significant. The statistical analyses were carried out using SPSS (version 27).

## 3. Results

### 3.1. Demographic Characteristics

The total number of participants in our survey was 11,294. Of all participants, 7520 participants were infected with SARS-CoV-2 and included in the final analysis. Around 46.0% were aged below 25 years. More than two-thirds of the study sample were females (69.5%). Approximately 6% of the study population reported diabetes, while around 4% reported that they had hypertension. Moreover, 971 (12.9%) of the study population were obese. Details of the study characteristics are listed in Table 1.

### 3.2. Characteristics of Acute COVID-19 among the Saudi Population

The characteristics of acute COVID-19 among participants are shown in Figure 1. Around 2886 (38%) of the study sample reported COVID-19 symptoms more than seven days since infection, followed by 4–6 days (2791 (37.1%)) and 1–3 days (1843 (24.5%)). In addition, more than 90% of the study sample had more than one symptom of COVID-19 disease, remarkably, with three acute symptoms, predominantly reported 1989 (26.4%). The most common medication used during acute infection was supplements (vitamins and minerals), 5652 (75.2%), suppurative care (analgesics and antipyretic), 5685 (75.6%), followed by oxygen therapy, 798 (10.6%), and steroids, 272 (2.6%). In addition, 6% (447) of our study participants needed hospitalization and, of these patients, around 50% stayed more than six days in the hospital. In addition, 118 (26%) of the hospitalized patients needed to be admitted to the ICU and mostly stayed 1–2 weeks, 38 (32.2%), followed by 1–3 days (32 (27.1%)) and more than two weeks (30 (25.4%)).

### 3.3. Characteristics and Risk Factors of Post-COVID-19 Conditions in the Saudi Population

The descriptive analysis of the characteristics of post-COVID-19 symptoms showed that 2737 (36%) of the study population had chronic symptoms after recovery from COVID-19 (Figure 2). This data showed the duration of post-COVID-19 conditions from 6 weeks to 6 months at 1063 (38.8%), followed by 4–6 weeks at 972 (35.5%). In our participant response regarding the number of post-COVID-19 symptoms, one symptom, 1287 (47.0%), was reported by most of the participants, followed by two symptoms, 919 (33.6%). In contrast, the most common symptoms that remained or appeared after recovery from COVID-19 were respiratory symptoms, 1835 (67.0%), nervous symptoms, 782 (28.6%) and psychological problems, 709 (25.9%), skin problems, 479 (17.5%), and other symptoms. Details of characteristics of post-COVID-19 symptoms are listed in Figure 2.

In the regression analysis (Table 2), age was a significant predictor of post−COVID−19 conditions, especially for those aged 60 years and above, 1.50 (1.13–1.99). Moreover, there was an association between gender variation and post−COVID-19 symptoms. For example, those who had more than one comorbid disease were associated with persistent post-COVID−19 symptoms, 1.66 (1.31–2.10), 1.59 (1.09–2.32), and 2.92 (1.60–5.35) for 1, 2, and more than four comorbid diseases, respectively. In addition, those who used chronic medications were associated with persistent COVID−19 symptoms.

Moreover, our data analysis showed that patient length of stay in hospital and ICU, duration of acute symptoms, and the number of acute symptoms is risk factors for developing post-COVID-19 symptoms (Table 3). In addition, the patient’s length of stay in hospital and ICU has time-dependent predictors of post-COVID-19 symptoms. Similarly, our data demonstrated that participants with more prolonged acute symptoms onset (7 days or more) had a 2.10 (1.91–2.32) fold risk of developing post-COVID-19 symptoms. Besides, participants who had more than one, four, and more than six acute symptoms were associated with persistent post-COVID-19 conditions 1.42 (1.27–1.59) and 12.61 (2.31–2.94), respectively.

## 4. Discussion

Regarding novel coronavirus outbreaks, available studies focused on describing the acute symptoms. However, studies on the community’s post-COVID-19 conditions are limited, especially in the Middle East. One previous study in the UK that coordinated analysis of survey data from 6907 individuals and 1.1 million individuals with COVID-19 diagnostic codes in electronic healthcare records data concluded that the proportion of long COVID-19 ranged from 7.8% to 17% [16]. In another prospective cohort study, in Russia, the authors included around 3358 adults and children with confirmed COVID-19 infection. The authors found that the prevalence of long COVID-19 was 50% and 20% among adults and children, respectively [17]. In our study, the prevalence of long COVID-19 was 34%, which was in the range of both studies. Post-COVID-19 conditions are delayed or long-term symptoms of COVID-19 beyond four weeks from the onset of acute symptoms [18]. Here, we described the occurrence of post-COVID-19 symptoms among the Saudi population. Post-COVID-19 symptoms have been previously described, which may occur due to the magnitude of virus-specific humoral and cellar immune responses [19], relapse or reinfection [20], exaggerated inflammatory response [21], and psychological elements, such as post-traumatic stress syndrome [22]. Most COVID-19 patients are symptomatic, and their acute symptoms may persist for up to three weeks [23,24]. Similarly, most of our patients (75%) were symptomatic and their symptoms continued for over three days. At the beginning of the COVID-19 pandemic, severe forms of COVID-19 were increasing and were associated with hospitalization in critical units. Then, after the development of COVID-19 vaccines, severe COVID-19 decreased significantly [25]. In our study, about 1.5% were admitted to the critical care unit. This low percentage could be due to the younger age of our population and the effectiveness of COVID-19 vaccines. In general, most respiratory viral infections are self-limited and resolve completely. However, some complications may occur and persist for a long time. Most of these are respiratory complications, such as post-infectious cough and airway hyperreactivity [26]. In addition, many papers recently reported post-COVID-19 symptoms and most were respiratory symptoms [14,15]. Likewise, most chronic symptoms in our study were due to respiratory problems (67%).

The socio-demographic data showed that most participants were under 40 years old, representing the general population of Saudi Arabia [27]. This study revealed that elderly age (>65 years old) predicts the probability of having a possible risk factor for post-COVID-19 conditions (Table 2), which was also in line with previous studies that reported that older age is a risk for developing long-COVID-19 symptoms [16]. However, it is well known that aging is associated with dysregulated immune responses, which may increase susceptibility or persistence to viral-related infection [28]. In addition, our study demonstrated that diabetes and hypertension are the major comorbidities among the Saudi population, which is a similar trend worldwide [29,30]. This report indicates that medication for treating chronic diseases may be linked to post-COVID-19 conditions. However, research is needed to confirm whether certain medicines may play a role in increasing susceptibility to SARS-Cov-2 infection. Long-term medication for hypertension and diabetes are the most prominent drugs used, which comply with the chronic disease burden in our study (Table 1). A previous study that analyzed the lung transcriptome sample from patients with comorbidities showed that ACE2 was highly expressed compared to the control group [31]. Moreover, ACE2 expression in patients with hypertension and diabetes is high due to the long-term medication of ACE inhibitors. A previous study on mice showed that ACE blockers increase the expression of ACE2 in the small intestine, lung, kidney, and brain; consequently, this may increase the risk of infection or reinfection, resulting in high virus titters [32]. In our study population, participants with one or more comorbidity diseases may be at increased risk of developing post-COVID-19 conditions. These results agree with potential risk factors previously identified in a multicenter observational study [33]. In addition, several studies reported comorbidities, including hypertension, diabetes, and asthma [16,17,34]. Therefore, the number of comorbidities matters in post-COVID-19 conditions. Gender variation has been linked to post-COVID-19 conditions, which has been described previously [14]. In our study, males appeared to have less risk of post-COVID-19 symptoms than females, similar to the previously reported data from hospitalized patients [14,17]. Moreover, the number and duration of acute COVID-19 symptoms are important risk factors in developing post-COVID-19 conditions. Limited studies investigated the relationship between the number and time of acute symptoms of SARS-CoV-2 infection with post-COVID-19 conditions.

Moreover, our data showed that length of stay (LOS) in hospital and ICU is essential in developing post-COVID-19 conditions, as previously reported [10,35,36]. Interestingly, the duration of LOS in hospitals and ICUs is a time-dependent predictor for post-COVID-19 conditions. Limitations of this study include insufficient collection of in-depth clinical information from participants during acute infection to investigate what type of acute infection is more associated with the possible risk of post-COVID-19 conditions. Moreover, our study did not monitor whether the comorbid complications changed or were aggravated during the post-COVID-19 period compared to the before-COVID-19 period. Therefore, extensive studies are needed to investigate the potential risk factors of post-COVID-19 conditions. Moreover, due to the cross-sectional nature of this report, insufficient data could limit the conclusion of our study. However, owing to the current epidemic and the nature of the objective of this study, we assume that we targeted a well-representative sample.

In conclusion, the prevalence of post-COVID-19 conditions among the Saudi population is 36%. Moreover, our study found that gender, age, number of chronic complications, long-term medication, LOS in hospital and ICU, and the duration of acute symptoms may play a role in post-COVID-19 conditions. Therefore, these findings should be taken with caution using the cross-sectional web-based-reported data. Moreover, healthcare professionals should be aware of post-COVID-19 symptoms in confirmed or suspected COVID-19 cases and the risk factors and long-term consequences are an *urgent* public-health *research priority.*

## Figures and Tables

**Figure 1 diseases-10-00055-f001:**
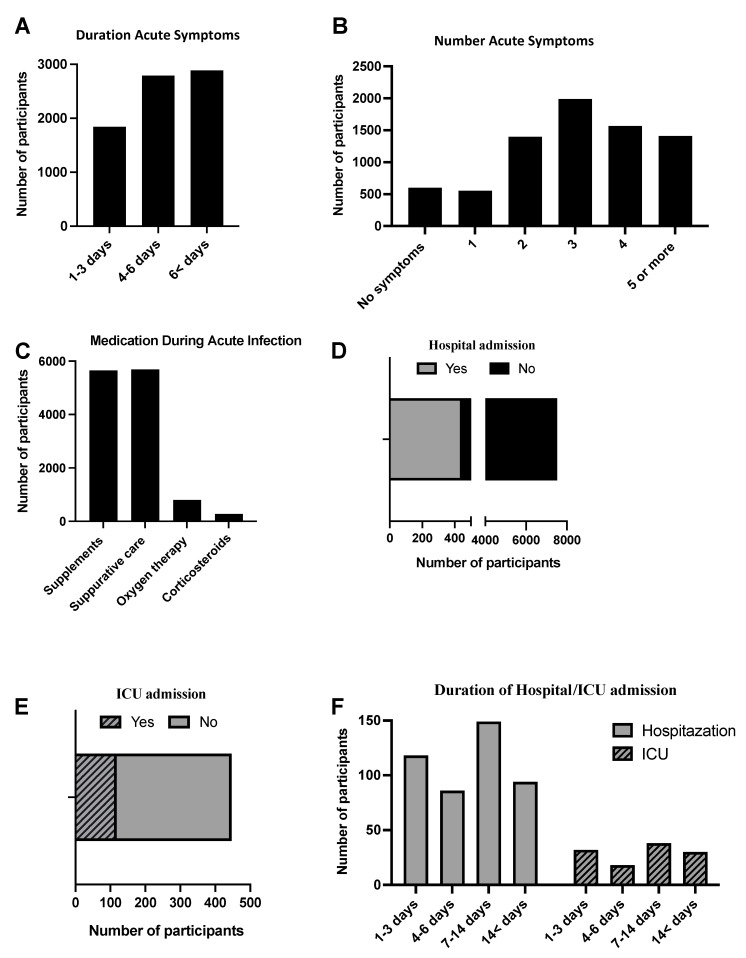
Descriptive analysis of acute SARS−CoV−2 among 7520 participants. (**A**,**B**) The data describe the duration and number of acute symptoms among participants. (**C**) The use of medication such as supplements, supportive care, steroids, and oxygen therapy during acute infection. (**D**) The rate of hospital admission among patients who were infected with SARS−CoV−2. (**E**) The rate of ICU admission among patients who were hospitalized during acute infection. (**F**) The duration of hospital and ICU stay during acute infection.

**Figure 2 diseases-10-00055-f002:**
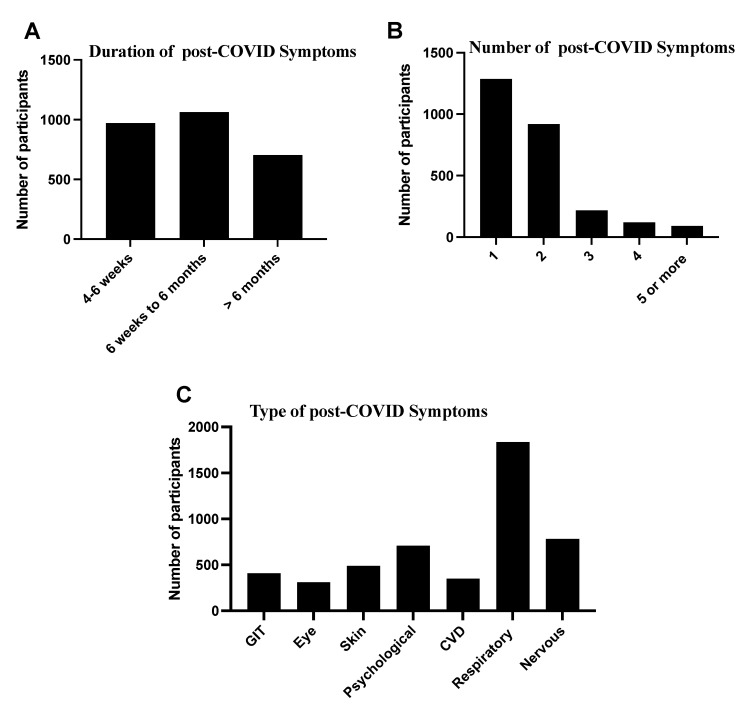
Characteristics of chronic or post−COVID−19 symptoms (*n* = 2737). (**A**,**B**) The data labelled the duration and number of chronic symptoms among the study’s participants. (**C**) The data described the most common type of chronic symptoms.

**Table 1 diseases-10-00055-t001:** Characteristics of study participants (*n* = 7520).

Variable	Frequency (%)
**Age**	
Below 25 years	3447 (45.8%)
25–40 years	2354 (31.3%)
41–59 years	1507 (20.0%)
60 years and above	212 (2.8%)
**Gender**	
Females	5230 (69.5%)
**BMI (Mean (SD) (kg/m^2^)**	24.0 (SD: 8.4)
**Smoking history**	
Non-smoker	6290 (83.6%)
Ex-smoker	351 (4.7%)
Smoker	879 (11.7%)
**Alcohol drinking (Yes)**	25 (0.3%)
**Number of comorbidities**	
0	6083 (80.9%)
1	975 (13.0%)
2	303 (4.0%)
3	111 (1.5%)
4 or more	48 (0.6%)
**Type of comorbidities**	
Diabetes mellitus	455 (6.1%)
Hypertension	298 (4.0%)
Liver diseases	5 (0.1%)
Cardiovascular diseases	68 (0.9%)
Chronic obstructive pulmonary diseases	193 (2.6%)
Immune diseases	77 (1.0%)
Psychological disorders	134 (1.8%)
Obesity	971 (12.9%)
Others	370 (4.9%)
**Chronic diseases medications use history (Yes)**	1060 (14.1%)
**Type of chronic diseases medications**	
Hypertension medication	205 (2.7%)
Oral antidiabetic	176 (2.3%)
Insulin	64 (0.9%)
Combination of oral antidiabetic and insulin	8 (0.1%)
Anti-hyperlipidemic agent	47 (0.6%)
Aspirin	23 (0.3%)
Thyroid disease medication	89 (1.2%)
Inhaled corticosteroids	29 (0.4%)
Gastro-protective agent	21 (0.3%)
Corticosteroids	19 (0.3%)
Antipsychotic agent	9 (0.1%)

**Table 2 diseases-10-00055-t002:** Predictors of persistent post-COVID-19 symptoms with demographic data (*n* = 2737).

Variable	Odds Ratio (95% CI)
**Age**	
Below 25 years (Reference category)	1.00
25–40 years	1.07 (0.96–1.18)
41–59 years	1.10 (0.98–1.24)
60 years and above	1.50 (1.13–1.99) **
**Gender**	
Females (Reference category)	1.00
Males	0.72 (0.65–0.80) ***
**BMI**	
24.0 kg/cm^2^ or lower (Reference category)	1.00
Above 24.0 kg/cm^2^	0.966 (0.86–1.06)
**Smoking history**	
Non-smoker (Reference category)	1.00
Ex-smoker	0.98 (0.78–1.23)
Smoker	1.00 (0.86–1.16)
**Number of comorbidities**	
0 (Reference category)	1.00
1	1.66 (1.31–2.10) ***
2	1.59 (1.09–2.32) *
3	-
4 or more	2.92 (1.60–5.35) **
**Type of comorbidities**	
Diabetes mellitus (Yes)	0.89 (0.66–1.19)
Hypertension (Yes)	0.81 (0.61–1.09)
Cardiovascular diseases (Yes)	0.81 (0.37–1.79)
Chronic obstructive pulmonary diseases (Yes)	0.95 (0.62–1.45)
Immune diseases (Yes)	1.00 (0.59–1.71)
Obesity (Yes)	0.66 (0.47–0.94) *
Psychological disorders (Yes)	0.88 (0.50–1.55)
**Chronic diseases medications use history**	
No (Reference category)	1.00
Yes	1.59 (1.39–1.82) ***

** p*-value ≤ 0.05, ** *p*-value ≤ 0.01, *** *p*-value ≤ 0.001.

**Table 3 diseases-10-00055-t003:** Predictors of persistent post-COVID-19 symptoms with acute SARS-CoV-2 (*n* = 2737).

**Length of hospital stay**	
One–three days (Reference category)	1.00
Four–six days	3.54 (1.33–9.43) *
Seven–fourteen days	2.08 (1.09–3.98) *
More than fourteen days	5.61 (2.24–14.07) ***
**Length of ICU stay**	
One–three days (Reference category)	1.00
Four–six days	3.54 (1.33–9.43) *
Seven days and above	3.02 (1.80–5.06) ***
**Duration of acute symptoms**	
1–3 days (Reference category)	1.00
4–6 days	0.82 (0.74–0.91) ***
7 days and above	2.10 (1.91–2.32) ***
**Number of acute symptoms**	
No symptoms (Reference category)	1.00
One	0.59 (0.48–0.72) ***
Two	0.67 (0.59–0.76) ***
Three	0.78 (0.70–0.88) ***
Four	1.42 (1.27–1.59) ***
Five	-
Six and above	2.61 (2.31–2.94) ***

* *p*-value ≤ 0.05, *** *p*-value ≤ 0.001.

## Data Availability

Data sharing does not apply to this article.

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
