# Peer review of "Determinants of Post-COVID-19 Conditions among SARS-CoV-2-Infected Patients in Saudi Arabia: A Web-Based Cross-Sectional Study"

_diseases, 2022, doi:10.3390/diseases10030055_

Round 1

Reviewer 1 Report

The article submitted by Samannodi et al conducted a cross-sectional, nationwide study on the prevalence of post-COVID complications using an online survey conducted in patients of Saudi Arabia. The study provided a rough picture of the incidence of post-COVID complications in patients. The study approach is great. However, we require more information on co morbid conditions they developed on post-COVID period. The information on whether the co-morbid complications arose during the post COVID- period or it got aggravated from previous condition. It would be great if the authors can explain the nature of chronic complications they experienced during post COVID period in detail which will be of great interest for the readers.

Minor comments

The authors should have avoided the innumerable number of cosmetic issues like spelling mistakes, spacing and others before submission.

Author Response

Manuscript diseases-1860286                                                   Makkah, 08/08/2022

Response to reviewers

We found the reviewers' comments very helpful and helped us to improve our manuscript further. We have addressed all reviewers' concerns in our point-to-point reply. For clarity, we have made some changes highlighted in yellow in the revised manuscript. All page numbers refer to the revised manuscript file with tracked changes included. We hope that our manuscript is now improved to a stage that it can be published in your journal.

Reviewers’ comments to the authors:
Reviewer #1):

The article submitted by Samannodi et al. conducted a cross-sectional, nationwide study on the prevalence of post-COVID complications using an online survey conducted in Saudi Arabian patients. The study provided a rough picture of the incidence of post-COVID complications in patients. The study approach is great. However, we require more information on co-morbid conditions they developed on post-COVID period. The information on whether the co-morbid complications arose during the post COVID- period or it got aggravated from previous condition. It would be great if the authors can explain the nature of chronic complications they experienced during post COVID period in detail which will be of great interest for the readers.

We thank the reviewer for pointing this out. It is an interesting point; however, we don’t have access to the participants to monitor the comorbidities before and during the COVID-19 period. This could be a great cohort for future study. We update the discussion to include some studies that discuss this information in line 200- 205. Besides, based on the reviewer's comment, we have now added this point to the limitations of our study.

Minor comments

The authors should have avoided the innumerable number of cosmetic issues like spelling mistakes, spacing and others before submission.

We thank the reviewer for the advice, and the manuscript was revised by native English speakers for any spelling and grammatical errors. 

Reviewer 2 Report

The manuscript entitled “Determinants of Post-COVID Conditions Among SARS-CoV-2 Infected Patients in Saudi Arabia: A Web-based Cross-sectional Study’’ submitted by Mohammed Samannodi et al, aims to investigate the incidence of post-COVID conditions among the general population of Saudi Arabia. Authors conducted a cross-sectional, nationwide study using an online survey conducted in Saudi Arabia from July to September 2021. They used various social media platforms using platforms such as Twitter, WhatsApp, and Facebook. They enrolled 7,520 individuals who had been infected with SARS-CoV-2. Most patients in this study were symptomatic, and their acute symptoms persisted for more than six days and amongst those, most of these complications were respiratory in origin, neurological, psychological, or skin-related

 They concluded that the incidence of post COVID conditions among the Saudi population was high, which urges further investigation into the risk factors associated with post COVID symptoms.

The authors have reported various studies, in the manuscript but there are certain things that need to be corrected and some needs the addition of literature.

General comments:

Ø  Authors have used different modes and deduced that the proportion of long-term complications reported in this study is 36% among SARS-CoV-2 infected individuals. In addition, female, old age, number of chronic complications, long-term medication, length of stay in hospital and intensive care unit, duration of acute symptoms may play significant predictors of post-COVID symptoms, but I wonder whether authors performed any invitro experiments to validate these findings.

Ø  I suggest, authors should write detailed legends in every figure to make it clear what were the criteria and how significant it is.

Ø  Please write the discussion section in detail.

Ø  Authors have made some figure, but I wonder the statistics was applied to the figures.

Ø  Authors should make comparisons in the patients with different type of symptoms and then compare those with the existing and non-existing ones.

Ø  Even I can not see and SD or significance on the bar graphs.

Author Response

Manuscript diseases-1860286                                                   Makkah, 08/08/2022

Response to reviewers

We found the reviewers' comments very helpful and helped us to improve our manuscript further. We have addressed all reviewers' concerns in our point-to-point reply. For clarity, we have made some changes highlighted in yellow in the revised manuscript. All page numbers refer to the revised manuscript file with tracked changes included. We hope that our manuscript is now improved to a stage that it can be published in your journal.

Reviewers’ comments to the authors

Reviewer #2):

Ø  Authors have used different modes and deduced that the proportion of long-term complications reported in this study is 36% among SARS-CoV-2 infected individuals. In addition, female, old age, number of chronic complications, long-term medication, length of stay in hospital and intensive care unit, duration of acute symptoms may play significant predictors of post-COVID symptoms, but I wonder whether authors performed any in vitro experiments to validate these findings.

We thank the reviewer for this comment, and it would be hard to apply the in vitro study to understand the predictors of post-COVID symptoms. However, it is possible to conduct the in vivo or ex vivo study on mice models such as humanized mouse models (MISTRG6-hACE2) of different ages and gender to evaluate if SARS-CoV-2 infection could aggravate the symptoms and consequently develop a sign from long COVID-19. Unfortunately, we don’t have access to this mouse model.

Ø  I suggest, authors should write detailed legends in every figure to make it clear what were the criteria and how significant it is.

Thanks for pointing this out; our purpose in generating the figures is to make them simple for the readers instead of the tables. We measured the association or significance in table 2 and table 3.  

Ø  Please write the discussion section in detail.

We update the discussion to include some studies that discuss this information in lines 200- 205

Ø  Authors have made some figure, but I wonder the statistics was applied to the figures.

As we claimed before, these figures had been generated to simplify the data result instead of the tables. The figures were generated to display the “count” of different categorical data.

Ø  Authors should make comparisons in the patients with different types of symptoms and then compare those with the existing and non-existing ones.

We thank the reviewers for this suggestion; however, when we built the questionnaire for this study, we didn’t include the type of symptoms as we were interested in describing the number (percentage) of symptoms and how these could be predictors for long-COVID-19. We include the questionnaire in the supplementary materials    

Ø  Even I can not see and SD or significance on the bar graphs.

The data were recorded as ordinal and not continuous; therefore we cannot calculate the SD. The graph measures the number of participants (counts); we have updated the figure for more clarification.   

Round 2

Reviewer 2 Report

The revise manuscript entitled “Determinants of Post-COVID Conditions Among SARS-CoV-2 Infected Patients in Saudi Arabia: A Web-based Cross-sectional Study’’ submitted by Mohammed Samannodi et al, aims to investigate the incidence of post-COVID conditions among the general population of Saudi Arabia. Authors conducted a cross-sectional, nationwide study using an online survey conducted in Saudi Arabia from July to September 2021. They used various social media platforms using platforms such as Twitter, WhatsApp, and Facebook. They enrolled 7,520 individuals who had been infected with SARS-CoV-2.

Authors have revised certain points and discussed few of them.

The authors have reported various studies, in the manuscript but there are certain things that need to be corrected and some needs the addition of literature.

Author Response

We thank the reviewer for careful reviewing our munscript. We have updated the introduction and the discussion as labelled in yellow the tracked version of the manuscript.